# Outcomes of Beta-Lactam Allergic and Non-Beta-Lactam Allergic Patients with Intra-Abdominal Infection: A Case–Control Study

**DOI:** 10.3390/antibiotics11121786

**Published:** 2022-12-09

**Authors:** Tayma Naciri, Boris Monnin, Alix Pantel, Claire Roger, Jean-Marie Kinowski, Paul Loubet, Jean-Philippe Lavigne, Albert Sotto, Romaric Larcher

**Affiliations:** 1Department of Infectious and Tropical Diseases, Nimes University Hospital, 30000 Nimes, France; 2VBIC (Bacterial Virulence and Chronic Infection), INSERM (French Institute of Health and Medical Research), Montpellier University, Department of Microbiology and Hospital Hygiene, Nimes University Hospital, 30000 Nimes, France; 3Anesthesiology and Critical Care Medicine, Nimes University Hospital, 30000 Nimes, France; 4Department of Pharmacy, Nimes University Hospital, 30000 Nimes, France; 5VBIC (Bacterial Virulence and Chronic Infection), INSERM (French Institute of Health and Medical Research), Montpellier University, Department of Infectious and Tropical Diseases, Nimes University Hospital, 30000 Nimes, France; 6PhyMedExp (Physiology and Experimental Medicine), INSERM (French Institute of Health and Medical Research), CNRS (French National Centre for Scientific Research), University of Montpellier, Department of Infectious and Tropical Diseases, Nimes University Hospital, 30000 Nimes, France

**Keywords:** beta-lactam, penicillin, allergy, intra-abdominal infection, outcomes

## Abstract

Background: In the case of intra-abdominal infections (IAI) in beta-lactam (BL) allergic patients, empiric antimicrobial therapy without BL is recommended; however, data regarding the outcome with alternative regimens are scarce. This study aimed to compare the outcomes of BL allergic (BLA) patients with IAI to those who were non-BLA (NBLA). Method: We conducted a case–control study in a French teaching hospital, between 1 January 2016 and 31 August 2021. BLA patients with IAI treated with fluoroquinolone or aztreonam and metronidazole were matched with controls treated with BL, on age, sex, disease severity, IAI localization, and healthcare-associated infection (HAI) status. We compared rates of therapeutic failures, adverse events, and HAI, and then assessed factors associated with therapeutic failure using a logistic regression model. Results: The therapeutic failure rate was 14% (*p* > 0.99) in both groups of 43 patients, and there was no significant difference in the adverse events rate (*p* > 0.99) and HAI rate (*p* = 0.154). Factors independently associated with therapeutic failure were higher BMI (OR 1.16; 95%CI [1.00–1.36]; *p* = 0.041), longer hospital length of stay (OR 1,20; 95%CI [1.08–1.41]; *p* = 0.006), and inadequate empiric antimicrobial therapy (OR 11.71; 95%CI [1.43–132.46]; *p* = 0.025). Conclusion: The outcomes of BLA patients with IAI treated without BL were the same as those for NBLA patients treated with BL.

## 1. Introduction

Intra-abdominal infections (IAI) are some of the most frequent abdominal emergencies and one of the first causes of septic shock [1]. A broad range of bacterial pathogens can be involved, originating mostly from the endogenous digestive flora [2]. Enterobacterales are the predominant species, followed by Gram-positive bacteria (streptococci and enterococci) and anaerobes. Importantly, most IAI are polymicrobial infections [3]. Consequently, several guidelines recommend the use of beta-lactams (BLs) as a first-line empiric therapy, usually in association with metronidazole and sometimes with vancomycin [4,5,6].

If a patient has a reported allergy to BL, the physician in charge should first take into account the over-reporting of BL allergy since only 1 to 20% of patients labelled as allergic have a confirmed allergy [7]. Then, the low (2%) cross-reactivity between penicillin and cephalosporin should be considered [8]. Nevertheless, in some cases, depending on the type of allergy, the entire BL class is avoided for IAI management. In these cases, guidelines recommend the use of alternative regimens based on fluoroquinolones, aztreonam or tigecycline. However, the use of alternative antibiotics in the case of BL allergy is known to increase the rate of surgical site infections (SSI) in a wide range of surgeries [9,10,11,12] and the rate of clinical failure in Gram-negative-bacilli-related infections [13]. Surprisingly, there is a lack of literature comparing the prognosis of patients with IAI treated with BL-based regimens to those treated with an alternative treatment due to a BL allergy. In particular, there are no data available for the last decade, whereas the prevalence of fluoroquinolone-resistant Enterobacterales has increased during this period [14].

Thus, this study aimed to assess the outcomes of beta-lactam allergic (BLA) patients with IAI treated without BL antibiotics to those of non-beta-lactam allergic (NBLA) patients treated with BL antibiotics, in a recent six-year period, in order to determine if beta-lactam allergy was associated to higher rates of therapeutic failure, adverse events, and healthcare-associated infections.

## 2. Results

The study inclusion process is depicted in Figure 1. During the study period, 6234 hospital stays were related to patients with IAI. We also identified 2209 hospital stays where the patient had received tigecycline, or an association of ciprofloxacin, ofloxacin or levofloxacin with metronidazole, or the association of aztreonam with metronidazole. Of them, we included 43 BLA patients in the analysis, and we matched them with 43 NBLA controls treated with BL.

### 2.1. Patients’ Characteristics

Among the 86 patients included in the analysis, 74% were female and the median age at admission was 66 years (interquartile range (IQR), 47–76). The two groups (BLA and NBLA) were well-balanced. Their demographic characteristics, detailed diagnoses, and characteristics of infection, including severity and inflammatory markers, are included in Table 1.

Most of the patients (48%) presented with biliary tract infections, followed with IAI affecting the appendix (23%), the colon (16%), and the gastroduodenal tract (2%). Eight additional patients (9%) had post-surgical peritonitis or peritoneal abscess at presentation. The initial surgeries were cholecystectomy (*N* = 2), appendectomy (*N* = 1), hysterectomy (*N* = 1), rectal et vesical prolapsus surgery (*N* = 1), colectomy (*N* = 1), bypass surgery (*N* = 1), and sleeve gastrectomy (*N* = 1). Some patients had extra-abdominal co-infections at admission, namely, a chronic hip prosthesis infection in one patient in the BLA group, and two urinary tract infections in the NBLA group. There were six healthcare-associated IAIs (14%) in both groups.

### 2.2. Isolated Pathogens, Antimicrobial Resistance, and Antibiotic Treatments

Fifty-seven patients (66%) underwent a source control procedure (Table 1). Of them, 43 (75%) had a microbiological sample collected. Among the whole population, 36 patients (42%) had at least one positive microbiological culture and more than 110 microorganisms were isolated (seven patients were reported to have a polymicrobial anaerobic flora). The documentation details and overview of administered treatments are reported in Table 2.

Six patients (14%) in the BLA group received aztreonam and metronidazole, including two patients without initial anti-Gram-positive coverage (of which, one later received vancomycin then linezolid due to *Enterococcus faecalis*-positive culture). Twenty-two patients (51%) in the BLA group received a fluoroquinolone with metronidazole, whereas eight patients (19%) treated with these antibiotics received at least one additional dose of gentamicin.

The details of inadequate empiric and directed antimicrobial therapies are summarized in Table 3.

### 2.3. Outcomes

The rates of therapeutic failures (*p* > 0.999) and adverse events attributable to antimicrobial therapy (*p* > 0.999) were the same in both groups. There was no significant difference (*p* = 0.3) in secondary healthcare-associated infections (HAIs). The reasons for therapeutic failures, type of HAI, and adverse events are reported in Table 3.

The Kaplan–Meier curves showing time to therapeutic failure for both groups are available in Figure 2. The probability of therapeutic success was 86%, 95%CI [76–97%] in both groups (log-rank test: *p* > 0.999).

The results of the robustness analysis, excluding the patients treated with aztreonam and those who received one dose of cefazolin during surgery, confirmed that BLA and NBLA patients have similar rates of therapeutic failures (*p* > 0.999), adverse events (*p* > 0.999), and secondary HAIs (*p* = 0.4) as detailed in Appendix A. Moreover, subgroup analyses highlighted that patients with therapeutic failure in both groups have comparable IAIs and disease severity (Appendix A).

In the multivariate analysis, factors associated with therapeutic failure were higher body mass index (BMI), odds ratio (OR) 1.16, 95% confidence interval (95%CI) [1.00–1.36], *p* = 0.041; longer hospital length of stay, OR 1.20 95%CI [1.08–1.41], *p* = 0.006; and inadequate empiric antimicrobial therapy, OR 11.71 95%CI [1.43–132.46], *p* = 0.025. Treatment with or without BL was not associated with therapeutic failure (Table 4).

## 3. Discussion

We reported herein the results of a recent case–control study aiming to compare the prognosis of BLA patients to NBLA patients with IAI. We found no significant difference between both groups in rates of therapeutic failures, adverse effects related to antibiotics, and healthcare-associated infections. In this retrospective study including 86 patients with a median Apache II score of 9, reflecting a non-negligible disease severity, we highlighted that a higher BMI, a higher hospital length of stay, and inadequate empiric antimicrobial therapy were factors independently associated with therapeutic failure.

The US and French guidelines for the management of IAI [4,5,6] recommend the use of BL-based regimens such as the combination of a third-generation cephalosporin and metronidazole or piperacillin/tazobactam with or without aminoglycoside (in the most severe forms). However, in the case of BL allergy, recommendations for antibiotic regimens have a maximum evidence grade of 2-B because the few studies comparing BL to alternative treatments are individual cohort studies and low-quality randomized controlled trials (RCT).

Our results are in agreement with those presented in a recent meta-analysis [15] pooling seven RCTs that reported the non-inferiority of fluoroquinolone-based regimens compared to BL-based regimens for complicated IAI [16,17,18,19,20,21,22]. Importantly, our study provides some recent data on the efficacy and safety of aztreonam-based regimens in IAI, which are otherwise lacking. Indeed, no RCT evaluating the use of this agent has been published since 1994 [23], and only one used a BL (imipenem–cilastatin) as a control against aztreonam with clindamycin [24].

In accordance with large retrospective studies [25,26,27,28], our results underlined the need for adequate empiric antimicrobial therapy to improve IAI outcomes. Empirical therapy for patients with IAI must include coverage against Enterobacterales and anaerobes, and in some cases, of *P. aeruginosa* and/or *Enterococcus* sp. [29].

Anaerobes coverage is reported to be a driver of IAI prognosis [30] and there is a consensus that it should always be part of IAI treatment [6]. Due to its good to excellent antimicrobial activity against Enterobacterales and most anaerobic microorganisms, moxifloxacin monotherapy has been proposed for patients with IAI [31]. However, a recent meta-analysis [15] suggested moxifloxacin was slightly inferior to BL, reigniting the debate on the use of fluoroquinolones in IAI. In our work, as in most clinical trials evaluating fluoroquinolones versus cephalosporins for IAI management [15], the anaerobes coverage is provided by the use of metronidazole, although its use is not mandatory in patients treated with piperacillin–tazobactam or imipenem [29].

In the same line, piperacillin–tazobactam and imipenem provide enterococcal coverage. In our case–control study, inappropriate empiric antibiotic therapy in nine out of the sixteen patients affected was due to non-coverage of *Enterococcus* sp. (in four BLA patients and five NBLA patients), partly because third-generation cephalosporins do not provide enterococcal coverage. Enterococci have been associated with higher morbidity and mortality, especially in the case of severe complicated IAI (cIAI) [32,33,34], but the empirical enterococcal coverage in lower-risk IAI did not improve treatment success in a recent meta-analysis of 23 RCT and 13 observational studies by Zhang et al. [35]. Nevertheless, in the case of severe IAI and BL allergy, anti-Gram-positive coverage with vancomycin remains recommended in all guidelines, in association with fluoroquinolones or aztreonam and metronidazole [4,5,6]. In our study, coverage of Gram-positive bacteria was absent for two patients treated with aztreonam, one of which experienced therapeutic failure.

Levofloxacin and moxifloxacin are fluoroquinolones with good activity against Gram-positive cocci, especially the latter, which has the lowest minimal inhibitory concentrations (MICs) for *Enterococcus* species [36]. However, data on its use in clinical practices for the treatment of enterococcal-related infections are scarce [37]. In addition, moxifloxacin has a higher risk of QTc interval prolongation compared to other fluoroquinolones [38]. Consequently, it was not used in our study, in which the tolerance of antibiotic therapies was good since only two non-severe adverse effect were reported: a linezolid-related cytopenia and a *Clostridioides difficile* infection.

The role of enterococci in the pathogenesis of polymicrobial infections is still debated [39]. Indeed, the morbidity and mortality associated with the presence of *Enterococcus* in IAI seem not to be influenced by antibiotic treatment but by the pro-inflammatory role of *Enterococcus* species [39]. Moreover, experimental data have showed that *Enterococcus* has developed a synergic relationship with other bacteria [40]. As a consequence, treatment of Gram-negative bacteria and anaerobes could be sufficient (i.e., third-generation cephalosporins or fluoroquinolone plus metronidazole) [4]. However, a regimen including vancomycin is mandatory in patients with MRSA carriage or in those at risk for ampicillin-resistant *Enterococcus* such as those with hepatobiliary diseases, hepatic transplantation, previous antimicrobial therapy, severe IAI and HA-IAI, and especially in BLA patients [32,41,42].

Among the three RCTs evaluating ciprofloxacin plus metronidazole [16,17,19], the most recent was published in 2006, which could limit the current interpretation of those trials considering the rising fluoroquinolone resistance in Enterobacterales worldwide. Indeed, in all European countries, the rate of *E. coli* strains resistant to fluoroquinolone has increased, such as in France where rates increased from 8.2% in 2004 to 14.8% in 2021 [14]. Nevertheless, our results suggested that fluoroquinolones remained an acceptable therapeutic option compared to third-generation cephalosporins, and resistance to third-generation cephalosporins and fluoroquinolones are at similar rates in our study (11.8% vs. 13%) as has been reported at a larger scale [14]. The French National Observatory of Bacterial Antibiotic Resistance also published some positive data showing that although all strains of *E. coli* in northern French hospitals were less sensitive to ciprofloxacin in 2018 (87% of sensitive strains) than in 2000 (95% of sensitive strains), there was a stabilization of resistance in the years 2015–2018 compared to the all-time low of 85% of *E. coli* ciprofloxacin-sensitive strains in 2011–2013. In the same report, *E. coli* sensitivity to cefotaxime fell from 100% in 2000 to 92% in 2018. [43]

Our study also identified higher BMI as an independent factor associated with therapeutic failure. Increased BMI is a well-known risk factor for surgical site infection [44] and subtherapeutic dosage of antimicrobials, especially in critically ill patients [45]. Allard et al. [46] suggested using a weight-adjusted dose to calculate a 4–5 mg/kg/dose of ciprofloxacin (up to 800 mg IV twice a day), but data are sparce for other fluoroquinolones. However, fluoroquinolones have good peritoneal and biliary penetration [47], which could limit the clinical impact of low plasma concentration. For critically ill patients with cIAI, optimizing antimicrobial therapy is challenging and some have suggested to use higher BL doses [48]. In this context, few data are available for aztreonam, but it could be used at the highest labeled dose of 2 g IV four times daily in BLA patients, particularly in the case of obesity [49]. Notably, aztreonam is active against *P. aeruginosa*; thus, this is an interesting therapeutic option in association with metronidazole and vancomycin for the treatment of healthcare-associated IAI.

Interestingly, aminoglycoside use was not significantly associated with therapeutic failure in the multivariate analysis. This could be explained, in part, by the variability in their indication in our study. Indeed, aminoglycosides have been used in severe IAI but also in nine (of 22) patients who received them as antibiotic prophylaxis. Our multivariate analysis also identified a longer hospital length of stay as a factor associated with therapeutic failure, which underlined the consequences of treatment failure. Patients needed to be reoperated on and treated with a broad-spectrum antibiotic, which prolonged their hospital length of stay and increased the costs associated to their stay [50].

The present study has several limitations. First, the small size of the cohort could limit the conclusions due to a lack of power to detect a difference in prognosis between groups. It could also have led to a misestimation of the importance of adequate source control, which has been reported by others as the factor most strongly associated with therapeutic success [3]. However, we tried to limit bias by matching controls according to the source control procedure. Second, the retrospective and monocentric design of the study could have introduced bias in data collection and limit result generalization. Nonetheless, the rates of therapeutic success in our cohort were consistent with those previously reported [15] as was the rate of healthcare-associated infections [51,52]. Third, we used the Apache II score to evaluate the severity of illness in patients who were not all hospitalized in ICU, which is the setting for which it was validated [53]. This could explain why we recorded a very low mortality rate at 1.2%, although the median Apache II scores in our cohort were in line with the RCT reporting the highest scores [16,17]. Finally, we included two patients who had a cefazolin dose per-operatively and patients treated with aztreonam in the BLA group. However, the results of the robustness analysis, excluding these patients, confirmed the results of the main analysis.

## 4. Materials and Methods

### 4.1. Study Design and Settings

We conducted an observational retrospective, case–control study in a French teaching hospital, between 1 January 2016 and 31 August 2021. During the study period, around 50,000 inpatients per years were admitted in our 1773-bed hospital (including 51 beds of critical care and 235 surgery beds).

### 4.2. Patients

All consecutive patients hospitalized between 1 January 2016 and 31 August 2021 in the hospital database were screened using the 10th International Classification of Diseases (ICD-10), codes K35 (acute appendicitis), K36 (other appendicitis), K37 (unspecified appendicitis), K57 (diverticular disease of intestine), K63.0 (abscess of intestine), K63.1 (non-traumatic perforation of intestine), K65 (peritonitis), K75.0 (abscess of liver), K80.0, K80.1, K80.3, K80.4 (calculus of gallbladder with cholecystitis, calculus of bile duct with cholangitis or cholecystitis), K81 (cholecystitis), and K83.0 (cholangitis).

This patient list was cross-referenced with the pharmacy’s record of all patients who had received an association of ciprofloxacin, ofloxacin, levofloxacin or aztreonam with metronidazole or clindamycin, or tigecycline, during the same time period. Then, we reviewed patient medical charts.

Clinical records were retrospectively reviewed by an internist (T.N.) and an ID physician (B.M.). Whenever a discrepancy between reviewers appeared, diagnosis was discussed with a third reviewer (R.L.) until a consensus was achieved.

Beta-lactam allergic patients (or patients treated as such, without BL) were included in the study (case group). Patients with peritonitis related to peritoneal dialysis catheters, and patients aged under 18 years old were excluded. Then, a control group of NBLA patients treated with BL were matched on the basis of age (+/− 2 years), gender, disease severity (Apache II score +/− 2 points), upper or lower digestive tract origin of infection, healthcare-associated nature of the infection, and, if possible, type of source control procedure, by extracting patients from the cohort of admissions to our hospital for IAI during the same time period.

### 4.3. Study Definitions

For the purpose of this study, aztreonam was categorized as a non-BL antibiotic because it has very low to no cross-reactivity with other BL [54,55] and because it is very often used in BLA patients in clinical practice.

If the microbiological culture of a sample was positive for a polymicrobial anaerobic flora, this flora was quoted as three species when counting the number of pathogens identified. When fluoroquinolones susceptibility testing was not performed on *Enterococcus* spp. or *Streptococcus* spp., they were classified as resistant. An antibiotic therapy was classified “adequate” if all pathogens identified were sensitive to at least one antibiotic prescribed to treat the IAI.

Prolonged corticosteroid therapy was defined as an intake of more than 10 mg of equivalent prednisone for over 14 days according to the French High Council of Public Health [56].

Healthcare-associated infections were defined according to the European Center for Disease Prevention and Control (ECDC) classification [57].

Intra-abdominal infections were classified as: appendicitis, cholecystitis, cholangitis, diverticulitis, intra-abdominal abscess, and peritonitis. A patient could have several diagnoses at once: for example, appendicular peritonitis was quoted as both appendicitis and peritonitis. Complicated IAI was defined as IAI with abscess or peritonitis according to the Infectious Disease Society of America (IDSA) classification [5].

The primary endpoint was clinical failure, defined according to the Food and Drugs Administration (FDA) recommendations for clinical drug trials for cIAI [58]. In short, clinical failure was defined as the occurrence of death (of any cause), SSI, unplanned surgical or percutaneous drainage procedures for complication or recurrence of IAI, or initiation of another antibiotic for worsening symptoms or signs of IAI, from start of the initial antibiotic therapy until day 28.

### 4.4. Data Collection

Demographical, clinical, and biological data were collected in the digital medical record for each patient.

In detail, the date of hospitalization and discharge, antimicrobial therapy type, and duration and the type of surgery, if any, was recorded, as well as the delay between diagnosis and invasive source control intervention, and the time to diagnosis after symptoms onset.

We recorded the microbiological documentation of the IAI, the number of cultured microorganisms, their resistance to antibiotics, the adequation of empirical and directed antimicrobial therapy. All microbiological results were retrospectively reviewed by a microbiologist (A.P.). We also recorded healthcare-associated infections apart from SSI, and adverse effects likely attributable to antibiotics.

Finally, we recorded the patients’ alcohol and tobacco use, history of abdominal surgery, body mass index (BMI), nature of BL allergy, the presence of immunodeficiency, maximum recorded temperature, maximum leukocytosis, maximum C-reactive protein (mg/L).

We evaluated comorbid conditions by calculating the Charlson index [59] for each patient and disease severity by calculating the Apache II score [53], at hospital or at ICU admission if the patient was secondarily transferred into such a unit.

### 4.5. Statistical Analysis

Data are described as median and interquartile range (IQR) or number and percentage as appropriate. We compared cases (BLA) and controls (NBLA). We analyzed variables associated with cases using conditional logistic regression on the pairs of cases and controls. To secure our results, we performed a robustness analysis by excluding cases who received an antibioprophylaxis with one dose of cefazolin and those treated with aztreonam, and their control. We generated survival curves using the Kaplan–Meier methodology and compared them by using a log-rank test. Then, we divided the study population in two groups (therapeutic failure and no therapeutic failure) and assessed factors associated with therapeutic failure using a logistic regression methodology. Variables with a *p*-value ≤ 0.1 in the univariate analysis were introduced into the multivariate analysis and were selected thereafter by using a backward selection method. We performed all statistical analyses with R software, version 4.2.0 (The R Foundation for Statistical Computing, Vienna, Austria). The Benjamini and Hochberg false discovery rate (FDR) correction method was used for *p*-value correction for multiple tests [60]. All tests were two-sided, and *p*-values less than 0.05 were considered statistically significant.

## 5. Conclusions

In this recent case–control study, we reported that patients with BLA treated for an IAI with alternative regimens, including a fluoroquinolone or aztreonam plus metronidazole, have the same outcomes as NBLA patients treated with a conventional BL-based regimen. None of the studied antibiotic regimens were associated with therapeutic failure, which strengthened our result, while we confirmed BMI, longer hospital length of stay, and inadequate antimicrobial therapy were independently associated with therapeutic failure.

Our results also underlined that in BLA patients with severe or healthcare-associated IAI, metronidazole should be associated with ciprofloxacin or aztreonam to ensure a coverage of *P. aeruginosa*, and with vancomycin or linezolid to treat streptococci and enterococci often involved in these settings. Further studies with larger numbers of patients are needed to confirm our results.

## Figures and Tables

**Figure 1 antibiotics-11-01786-f001:**
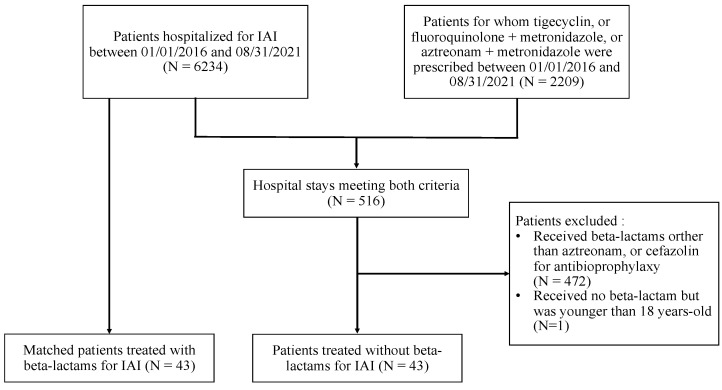
Flow chart of the study population.

**Figure 2 antibiotics-11-01786-f002:**
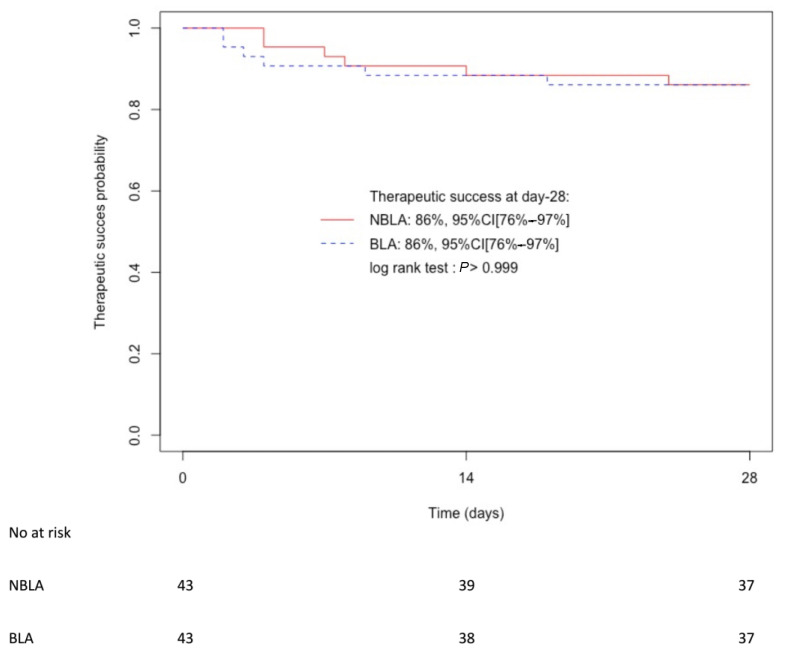
Kaplan–Meier curves of therapeutic success probability in non-beta-lactam allergic (NBLA) patients and beta-lactam allergic (BLA) patients. CI: confidence interval.

**Table 1 antibiotics-11-01786-t001:** Characteristics of the study population.

Characteristics	BLA (*N* = 43) *N* (%) or Median (IQR)	NBLA (*N* = 43) *N* (%) or Median (IQR)
Women	32 (74%)	32 (74%)
Median age, years	65 (47–76)	66 (47–76)
Alcohol > 20 g/day	2 (5%)	1 (2%)
Tobacco use > 4 unit/day	8 (19%)	3 (7%)
Charlson index	2 (1–7)	3 (1–6)
History of abdominal surgery	25 (58%)	29 (67%)
Corticosteroid treatment	2 (5%)	0 (0%)
Other immunodeficiency	1 (2%)	4 (9%)
BMI (kg/m^2^)	25.56 (23.52–29.59)	25.36 (22.75–29.27)
Type of BL allergy		
- Unspecified rash	3 (7%)	-
- Urticaria	6 (14%)	-
- Angio-oedema	13 (30%)	-
- Grade III anaphylaxis	1 (2%)	-
- Toxidermy	4 (9%)	-
- Unknown	16 (37%)	-
Type of infection		
- Upper digestive tract	21 (49%)	21 (49%)
- Cholangitis	11 (26%)	11 (26%)
- Cholecystitis	9 (21%)	12 (28%)
- Appendicitis	10 (23%)	10 (23%)
- Diverticulitis	7 (16%)	5 (12%)
Complicated IAI:	19 (44%)	19 (44%)
- Peritonitis	10 (23%)	15 (35%)
- Intra-abdominal abscess	9 (21%)	4 (9%)
HLOS (days)	5 (3–9.5)	7 (4–11)
Maximal temperature (°C)	37.3 (37–38.3)	37.5 (31–38.6)
Maximal CRP (mg/L)	174.5 (130.2–241.1)	192 (97.1–333.2)
Maximal leukocytosis (G/L)	14.22 (11.21–19.11)	16.73 (12.88–21.82)
Apache II score	9 (6–13)	9 (6–14)
Healthcare-associated IAI	6 (14%)	6 (14%)
Co-infection at admission	1 (2%)	2 (5%)
Diagnostic delay (days)	1 (1–4)	2 (1–3)
Source control delay (days)	2 (1–5)	3 (2–6)
Source control procedure	26 (60%)	31 (72%)
- Laparotomy	6 (14%)	7 (16%)
- Laparoscopy	15 (35%)	15 (35%)
- Percutaneous drainage	2 (5%)	4 (9%)
- ERCP	3 (7%)	5 (12%)
Antimicrobial therapy	43 (100%)	43 (100%)
- fluoroquinolones	42 (98%)	0 (0%)
- aztreonam	6 (14%)	0 (0%)
- ceftriaxone	0 (0%)	21 (49%)
- cefotaxime	0 (0%)	20 (47%)
- piperacillin–tazobactam	0 (0%)	9 (21%)
- amoxicillin–clavulanic acid	0 (0%)	18 (42%)
- metronidazole	43 (100%)	36 (84%)
- aminoglycoside	12 (28%)	10 (23%)
- glycopeptide	5 (12%)	6 (14%)
- antifungal agent	0 (0%)	3 (7%) ^1^
- duration (days)	10 (7–14)	10 (7–13)

^1^ Fluconazole (*N* = 2), caspofungin (*N* = 1). BLA: beta-lactam allergic, CRP: C-reactive protein, ERCP: endoscopic retrograde cholangiopancreatography, HLOS: hospital length of stay, IAI: intra-abdominal infection, IQR: interquartile range, NBLA: non-beta-lactam allergic.

**Table 2 antibiotics-11-01786-t002:** Documented cultures, antibiotic resistance, and antibiotics administered.

Characteristics	BLA (*N* = 43) *N* (%) or Median (IQR)	NBLA (*N* = 43) *N* (%) or Median (IQR)	*p*-Value
Surgical samples collected	17 (40%)	26 (60%)	0.054
Bloodstream infection	2 (5%)	6 (14%)	0.156
Patients with positive cultures	14 (33%)	22 (51%)	0.082
Gram-positive cocci	9 (21%)	11 (26%)	0.610
- *Enterococcus* spp.	6 (14%)	5 (12%)	0.747
- *Streptococcus* spp.	4 (9%)	7 (16%)	0.338
Gram-negative bacilli	13 (30%)	18 (42%)	0.263
- *Escherichia coli*	10 (23%)	14 (33%)	0.338
- *Proteus* spp.	2 (5%)	1 (2%)	0.564
- *Klebsiella pneumoniae* and *K. oxytoca*	3 (7%)	4 (9%)	0.694
- *Enterobacter* spp.	1 (2%)	1 (2%)	>0.999
- *Citrobacter* spp.	1 (2%)	1 (2%)	>0.999
- *Pseudomonas aeruginosa*	2 (5%)	0 (0%)	0.992
Anaerobes	8 (19%)	10 (23%)	0.597
- *Bacteroides* spp.	6 (14%)	3 (7%)	0.299
*Candida* spp.	0 (0%)	3 (7%)	0.990
Antimicrobial resistance in Enterobacterales			
3GC-resistant	0/17 (0%)	3/23 (13%)	
FQ-resistant	2/17 (12%)	3/23 (13%)	
Total	2/17 (12%)	6/23 (26%)	
Antimicrobial therapy duration (days)	10 (7–14)	10 (7–13)	0.847
Fluoroquinolones	42 (98%)	0 (0%)	
- ofloxacin	33 (77%)	0 (0%)	
- ciprofloxacin	8 (19%)	0 (0%)	
- levofloxacin	7 (16%)	0 (0%)	
Metronidazole	43 (100%)	36 (84%)	0.991
Aztreonam	6 (14%)	0 (0%)	0.991
Ceftriaxone	0 (0%)	21 (49%)	0.989
Cefotaxime	0 (0%)	20 (47%)	0.990
Piperacillin–Tazobactam	0 (0%)	9 (21%)	0.990
Amoxicillin–Clavulanic Acid	0 (0%)	18 (42%)	0.990
Amoxicillin	0 (0%)	1 (2%)	0.991
Aminoglycoside	12 (28%)	10 (23%)	0.621
Glycopeptide	5 (12%)	6 (14%)	0.747
Clindamycin	4 (9%)	0 (0%)	0.989
Antifungal agent	0 (0%)	3 (7%) ^1^	0.990

^1^ Fluconazole (*N* = 2), caspofungin (*N* = 1). BLA: beta-lactam allergic, NBLA: non-beta-lactam allergic, 3GC: third-generation cephalosporin, FQ: fluoroquinolone.

**Table 3 antibiotics-11-01786-t003:** Inadequate antimicrobial therapy, therapeutic failures, healthcare-associated infections, and antibiotic-related adverse events.

Characteristics	BLA (*N* = 43) *N* (%)	NBLA (*N* = 43) *N* (%)	*p*-Value
	Inadequate antibiotic therapy		
Empiric antimicrobial therapy	8 (19%)	8 (19%)	>0.999
Directed antimicrobial therapy	5 (12%)	1 (2%)	0.2
Details of inadequate antimicrobial therapy: empiric → directed			
*Enterococcus* sp. or *Streptococcus* sp.	7 → 4	5 → 0	
*Pseudomonas aeruginosa*	1 → 0	0 → 0	
Fluoroquinolone-resistant Enterobacterales	1 → 1	0 → 0	
3rd generation Cephalosporin resistant Enterobacterales	0 → 0	1 → 0	
Piperacillin–tazobactam-resistant Enterobacterales	0 → 0	2 → 1	
Methicillin-resistant *Staphylococcus epidermidis*	0 → 0	1 → 0	
Metronidazole-resistant *Bacteroides thetaiotaomicron*	1 → 1	0 → 0	
	Therapeutic failure		
	6 (14%)	6 (14%)	>0.999
Death	-	1	
Surgical site infection	-	1	
Unplanned surgery due to complication or recurrence of IAI	5	4	
Initiation of another antibiotic for worsening symptoms of IAI	2	4	
	Healthcare-associated infection		
	2 (5%)	6(14%)	0.3
Candidemia	1 ^1^	-	
Recurrent cholangitis	1	-	
Cystitis	1 ^1^		
Surgical site infection	-	1	
Pulmonary infection	-	2 ^2^	
Bloodstream infection	-	2	
*Clostridioides difficile* infection		1	
	Adverse event due to antibiotics		
	1 (2%)	1 (2%)	>0.999
Thrombocytopenia	1 ^3^	-	
*Clostridioides difficile* infection	-	1	

^1^ Both events happened in the same patient. ^2^ Hospital-acquired SARS-CoV-2 pneumonia causing the death of the patient (*N* = 1), empyema (*N* = 1). ^3^ Antibiotics administered: aztreonam, ofloxacin, metronidazole, and linezolid, the latter being the most likely culprit. BLA: beta-lactam allergic, IAI: intra-abdominal infection, NBLA: non-beta-lactam allergic.

**Table 4 antibiotics-11-01786-t004:** Univariate and multivariate analysis of factors associated with therapeutic failure.

Characteristics	Univariate OR (95% CI)	*p*-Value	Multivariate OR (95% CI)	*p*-Value
Inadequate empiric antimicrobial therapy	9.86 (2.27–44.76)	0.002	11.71 (1.43–132.46)	0.025
Aminoglycosides	5.51 (1.55–21.04)	0.009	2.89 (0.39–23.96)	0.293
BMI (per kg/m^2^)	1.12 (1.01–1.26)	0.028	1.16 (1.00–1.36)	0.041
HLOS (per day)	1.20 (1.10–1.35)	<0.001	1.20 (1.08–1.41)	0.006

BMI: body mass index, CI: confidence interval, HLOS: hospital length of stay.

## Data Availability

The authors consent to sharing the collected data with others. The raw data supporting the conclusions of this article will be made available by the authors, without undue reservation. Data will be available immediately after the main publication and indefinitely.

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
