# Peer review of "Outcomes of Beta-Lactam Allergic and Non-Beta-Lactam Allergic Patients with Intra-Abdominal Infection: A Case–Control Study"

_antibiotics, 2022, doi:10.3390/antibiotics11121786_

Round 1

Reviewer 1 Report

Interesting, but some points need to be revised:

- Lines 63-65. "Thus, this study aimed to assess the outcomes of beta-lactam allergic (BLA) patients 63 with IAI treated without BL antibiotics to those of non-beta-lactam allergic (NBLA) pa- 64 tients treated with BL antibiotics, in a recent 6-year period" For which reason? What is the primary objective of this paper?

- Lines 107-109. "without initial anti-gram-positive coverage (of which, one later received 106 vancomycin then linezolid due to Enterococcus faecalis-positive culture)." Authors underlined the use of vancomycin in the conclusion section for this disease, but they should discuss more.

- Lines 192-195. Authors should discuss about the abdominal and extra-abdominal textiloma in relation to antibiotics. Look at these papers:  -- doi: 10.1007/s00595-007-3654-x  --  doi: 10.1159/000505233.

- Table 4 must be discussed more in the text. Aminoglycosides seem not to affect therapeutic failure. Revised and discuss more.

- Lines 197-200. "Nevertheless, our results.. similar rates in our study (11.8% vs 13%) as it has been reported 200 at a larger scale [14]." Report more  previous studies and discuss them more.

- Lines 136-137. "Figure 1. Kaplan-Meier curves of therapeutic success probability in non-beta-lactam allergic (NBLA) patients and beta-lactam allergic (BLA) patients." This is probably Figure 2. Revise it.

Author Response

Interesting, but some points need to be revised:

First, we thank the reviewer for their interest to this work and for their thorough comments.

- Lines 63-65. "Thus, this study aimed to assess the outcomes of beta-lactam allergic (BLA) patients with IAI treated without BL antibiotics to those of non-beta-lactam allergic (NBLA) patients treated with BL antibiotics, in a recent 6-year period" For which reason? What is the primary objective of this paper?

The main objective of this work was to determine if beta-lactam allergy is associated to higher rates of therapeutic failure, adverse events and healthcare-associated infections, in patients with IAI. The recent 6-year period (2016-2021) has been chosen since it followed the last French guidelines for the management of IAI, which were updated in April, 2015. Moreover, it followed the last French Action Plan against Antimicrobial Resistance (2011-2016). As requested by the reviewer, we amended the final paragraph of the introduction to better emphasized the primary objective of paper. Please see lines 63-67, p2.

- Lines 107-109. "without initial anti-gram-positive coverage (of which, one later received vancomycin then linezolid due to Enterococcus faecalis-positive culture)." Authors underlined the use of vancomycin in the conclusion section for this disease, but they should discuss more.

As requested by the reviewer, we added a paragraph in the discussion, focusing on vancomycin use, particularly in patients with BLA, with MRSA carriage and in patients at risk for ampicillin-resistant Enterococcus such as patients with hepatobiliary diseases, hepatic transplantation, previous antimicrobial therapy, severe IAI and HA-IAI. Please see lines 204-214, p9.

- Lines 192-195. Authors should discuss about the abdominal and extra-abdominal textiloma in relation to antibiotics. Look at these papers:  -- doi: 10.1007/s00595-007-3654-x  --  doi: 10.1159/000505233.

We completely agree with the reviewer, textiloma is a particularly relevant issue for surgeons and ID physicians as it can mimic abscess or tumor and is an under-reported occurrence. However, none of the patients of our cohort had textiloma, and consequently we do not discuss this point in the present paper.

- Table 4 must be discussed more in the text. Aminoglycosides seem not to affect therapeutic failure. Revised and discuss more.

As correctly pointed by the reviewer, aminoglycoside use was not significantly associated with therapeutic failure in the multivariate analysis. This could be explained by the variability of their indication in our study. Indeed, aminoglycosides have been used in severe IAI (N=13) but also in nine patients which received them as antibiotic prophylaxis (N=5 in the BLA group, N=4 in NBLA group). Among the 22 patients treated with aminoglycosides, seven had therapeutic failure and almost all (but one) had a severe IAI. We added a statement in the discussion section (lines 255-258, p10). Globally, we also revised and discussed more the results of the table 4 (lines 258-262, p9).

- Lines 197-200. "Nevertheless, our results. similar rates in our study (11.8% vs 13%) as it has been reported 200 at a larger scale [14]." Report more previous studies and discuss them more.

We have amended the text in order to provide a better insight of fluoroquinolone and third generation cephalosporins in our country through the years, in order to highlight the similar resistance rates between those two antibiotics, and the increase of resistance compared to the 2000s, which is the period during which most trials comparing fluoroquinolones to beta-lactams in IAI were published.

- Lines 136-137. "Figure 1. Kaplan-Meier curves of therapeutic success probability in non-beta-lactam allergic (NBLA) patients and beta-lactam allergic (BLA) patients." This is probably Figure 2. Revise it.

Thanks to the reviewer for highlighting this mistake. The text has been revised.

Reviewer 2 Report

The authors reported the results of a case-control study aiming to compare the prognosis of BLA patients to NBLA patients with IAI. They found no significant difference between both groups in rates of therapeutic failures, adverse effects related to antibiotics, and healthcare-associated infections. However, the homogeneity of different infections is needed to be fully considered and interpreted. Moreover, the subgroups analysis may be needded for different severities of infections.

Author Response

The authors reported the results of a case-control study aiming to compare the prognosis of BLA patients to NBLA patients with IAI. They found no significant difference between both groups in rates of therapeutic failures, adverse effects related to antibiotics, and healthcare-associated infections. However, the homogeneity of different infections is needed to be fully considered and interpreted. Moreover, the subgroups analysis may be needed for different severities of infections.

We thank the reviewer for their helpful observations.

We definitely agree that the type of IAI determines the prognosis. To compare homogeneous populations, we matched the BLA group with a control group of NBLA patients based on age (+/- 2 years), gender, disease severity (Apache II score +/- 2 points), the type of infection and its nosocomial nature. Due to the large number of criteria used, we must acknowledge that we were forced to match two patients with a diverticulitis in the BLA group with two patients in the NBLA group with cholecystitis (see table 1). However, both infections have a good prognosis and none of the four patients had therapeutic failure.

As suggested by the reviewer, we performed a subgroups analysis for different infections and severities of infections (see table S2).

Reviewer 3 Report

This article by Tayma Naciri et al. describes an interesting study on the impact of therapeutic modifications induced by betalactam allergy on the management of intra-abdominal infection.

This manuscript deserves to be reviewed before possible publication.

Global: italicize "et al."; "vs."; prefer passive forms.

Table 1 does not require a p-value, as it is only a choice of controls, which does not require statistical tests.

Numbers less than 12 should be written out in full.

How did the authors take into account the risk inherent in the multiplicity of statistical tests?

Table 3: "1a" but what is the "a"?

As the authors stated, the inclusion of patients with cefazolin per operatively could be a bias easily overcome, by redoing the analysis after exclusion (even if the number is small it is to be taken into account). Ditto for aztreonam (a robustness analysis may help to maintain these patients...).

How were the number of subjects and the inclusion period determined?

How many authors took over the clinical records?

Did the authors contact a privacy committee? In the context, the ethics committee is not enough.

Data availability statement. It is not acceptable that the authors put so much condition to share their data. It is necessary that they put them on a free access bank.

Author Response

This article by Tayma Naciri et al. describes an interesting study on the impact of therapeutic modifications induced by betalactam allergy on the management of intra-abdominal infection.

This manuscript deserves to be reviewed before possible publication.

The authors thank the reviewer for their interest in our work and for their remarks.

Table 1 does not require a p-value, as it is only a choice of controls, which does not require statistical tests.

How did the authors take into account the risk inherent in the multiplicity of statistical tests?

As requested by the reviewer the p-values have been removed from table 1 and 2 which summarized the characteristics of groups BLA and NBLA. The Benjamini & Hochberg methodology have been used for p-value correction for multiple tests and p-values have been modified in table 4 and text (see methods and results sections).

Global: italicize "et al."; "vs."; prefer passive forms. Numbers less than 12 should be written out in full.

Thanks to the reviewer for these remarks. We revised the manuscript in accordance with the reviewer’s comments.

Table 3: "1a" but what is the "a"?

Thanks to the reviewer for highlighting this mistake. The table 3 has been modified to read “11”.

As the authors stated, the inclusion of patients with cefazolin per operatively could be a bias easily overcome, by redoing the analysis after exclusion (even if the number is small it is to be taken into account). Ditto for aztreonam (a robustness analysis may help to maintain these patients...).

We agree with the reviewer. We chose to include cefazolin prophylaxis because it is a widely used antibiotic even for patient who have an otherwise documented penicillin allergy, because of its low cross-reactivity with penicillin, and as such, we felt that those two patients reflected the real-life management of IAI in BLA patients. Importantly, one of those two patients accounted for a therapeutic failure, and the other was treated with aztreonam. As recommended by the reviewer we performed a robustness analysis by excluding patients treated with cefazolin and/or aztreonam (and their control). Please see methods, results, discussion (limits paragraph) and table S1.  

How were the number of subjects and the inclusion period determined?

The start of the inclusion period was chosen because French guidelines for the management of IAI, particularly in the context of beta-lactam allergy, were updated in April, 2015. Thus, we felt that the start of the year 2016 was a reasonable timepoint at which you could expect physicians to be aware of these guidelines and to adhere to them. It also followed the last French Action Plan against Antimicrobial Resistance (2011-2016). We conducted a retrospective analysis and found out that only 43 patients didn’t receive beta-lactams (apart from aztreonam) for IAI in the aforementioned time-period, after checking every patient file, and matched them one for one with controls. Obviously, the small size of the cohort could limit the conclusions due to a lack of power to detect a difference in prognosis between groups as we stated in the limit paragraph of the discussion. Taking into account the 14% rate of therapeutic failure, the inclusion of 43 patients in each group could detect an odd ratio ≥5 (Power = 90%, alpha risk = 5%).

How many authors took over the clinical records?

All the clinical records were retrospectively reviewed by an internist (T.N.) and an ID physician (B.M.). Whenever a discrepancy between reviewers appeared, diagnosis was discussed with an ICU physician (R.L.) until a consensus was achieved. All microbiological results were also retrospectively reviewed by a microbiologist (A.P.). A statement has been added in the method section.

Did the authors contact a privacy committee? In the context, the ethics committee is not enough.

We thank the reviewer for this cautious approach towards patient’s rights. This study is a retrospective, non-interventional study which has no influence on the patient’s care, and in accordance with French legislation, it only required an approval by an institutional review board and does not require a patient’s signed informed consent, or a privacy committee since the data collected are anonymized.

Data availability statement. It is not acceptable that the authors put so much condition to share their data. It is necessary that they put them on a free access bank.

As requested by the reviewer, we have changed the data availability statement to facilitate the sharing of the data that supports the conclusion of our work.

Round 2

Reviewer 2 Report

 The outcomes of BL allergic (BLA) patients with IAI and those non-BLA (NBLA) should also be compared between Gram-positive infection subgroups and Gram-negative infeciton subgroups,respectively.

Author Response

The outcomes of BL allergic (BLA) patients with IAI and those non-BLA (NBLA) should also be compared between Gram-positive infection subgroups and Gram-negative infection subgroups respectively.

As correctly pointed by the reviewer, the isolation of Gram-positive/negative bacteria could change the prognosis of patients with IAI, especially in BLA patients. These factors were associated with therapeutic failure in the univariate analysis (p = 0.03 and 0.02, respectively), thus, they have been included in the multivariate analysis, but remained not associated with therapeutic failure after adjustment on other cofactors. As reported in the Table 2, globally more patients had positive culture in the NBLA group (N=22, 51%) than in the BLA group (N=14, 33%). Consequently, in patient with therapeutic failure, more patients had positive culture for both Gram-positive bacteria (4 vs 2) and Gram-negative bacteria (6 vs 2) in the NBLA group. We amended the Table S2 accordingly.

Reviewer 3 Report

The manuscript has been revised according to my previous comments.

Author Response

The manuscript has been revised according to my previous comments.

The authors thank the reviewer for their constructive contributions.